# Neuroblastoma signalling models unveil combination therapies targeting feedback-mediated resistance

**Mathurin Dorel**[1,2], **Bertram Klinger**[1,2,5,6], **Tommaso Mari**[8], **Joern Toedling**[3], **Eric Blanc**[4], **Clemens Messerschmidt**[4], **Michal Nadler-Holly**[8], **Matthias Ziehm**[8], **Anja Sieber**[1,2,5], **Falk Hertwig**[3], **Dieter Beule**[4], **Angelika Eggert**[3,5,6,7], **Johannes H. Schulte**[3,5,6,7], **Matthias Selbach**[8], **Nils Blüthgen**[1,2,4,6,7]*

**1** Institute of Pathology, Charité-Universitätsmedizin Berlin, Berlin, Germany, **2** Integrative Research Institute for the Life Sciences and Institute for Theoretical Biology, Humboldt-Universität zu Berlin, Berlin, Germany, **3** Department of Pediatric, Division of Oncology and Haematology, Charité-Universitätsmedizin Berlin, Berlin, Germany, **4** Berlin Institute of Health, Berlin, Germany, **5** German Cancer Consortium (DKTK), partner site Berlin, Germany, **6** German Cancer Research Center (DKFZ), Heidelberg, Germany, **7** Berlin Institute of Health (BIH), Berlin, Germany, **8** Max Delbrück Center for Molecular Medicine, Berlin, Germany

* nils.bluethgen@charite.de

**Data Availability Statement:** The data and analysis packages are available at these repositories: RNA-Seq data: ENA PRJEB40670 (https://www.ebi.ac.uk/ena/browser/view/PRJEB40670) STASNet

## Abstract

Very high risk neuroblastoma is characterised by increased MAPK signalling, and targeting MAPK signalling is a promising therapeutic strategy. We used a deeply characterised panel of neuroblastoma cell lines and found that the sensitivity to MEK inhibitors varied drastically between these cell lines. By generating quantitative perturbation data and mathematical modelling, we determined potential resistance mechanisms. We found that negative feedbacks within MAPK signalling and via the IGF receptor mediate re-activation of MAPK signalling upon treatment in resistant cell lines. By using cell-line specific models, we predict that combinations of MEK inhibitors with RAF or IGFR inhibitors can overcome resistance, and tested these predictions experimentally. In addition, phospho-proteomic profiling confirmed the cell-specific feedback effects and synergy of MEK and IGFR targeted treatment. Our study shows that a quantitative understanding of signalling and feedback mechanisms facilitated by models can help to develop and optimise therapeutic strategies. Our findings should be considered for the planning of future clinical trials introducing MEKi in the treatment of neuroblastoma.

## Author summary

Only few targeted therapies are currently available to treat high-risk neuroblastoma. To address this issue we characterized the drug response of high risk neuroblastoma cell lines and correlated it with genomic and transcriptomic data. Particularly for MEK inhibition, we saw that our panel could be nicely separated into two groups of resistant and sensitive cell lines. Genomic and transcriptomic markers alone did not help to discriminate between responders and non-responders. We used signalling perturbation data to build

package: GitHub (https://github.com/molsysbio/
STASNet/releases/tag/Dorel2020)
Phosphoproteomics: https://itbgit.biologie.hu-
berlin.de/dorel/phosphoproteomics_tnb_
perturbations.

**Funding:** We acknowledge funding from the Berlin
Institute of Health (CRG1 Terminate NB) and from
the Federal Ministry of Education and Research
/BMBF/ (grant MSTARS, to NB and MS). The
funders had no role in study design, data collection
and analysis, decision to publish, or preparation of
the manuscript.

**Competing interests:** The authors have declared
that no competing interests exist.

cell line specific signalling models. Our models suggest that negative feedbacks within
MAPK signalling lead to a stronger reactivation of MEK in MEKi resistant cell lines after
MEK inhibition. Model analysis suggested that combining MEK inhibition with IGF1R or
RAF inhibition could be an effective treatment and we characterised this combination
using phosphoproteomics by mass-spectrometry and growth assays. Our study confirms
the importance of quantitative understanding of signalling and may help plan future clini-
cal trials involving MEK inhibition for the treatment of neuroblastoma.

## Introduction

Neuroblastoma is the most common and devastating extracranial childhood solid tumour,
accounting for 15% of all childhood cancer deaths. The 5-year survival rate is 75% overall, but
it is below 45% for so-called high-risk neuroblastoma that represent about 40% of patients [1–
3]. Telomere maintenance is a central hallmark of high-risk neuroblastoma [4], and approxi-
mately 50% of high-risk neuroblastoma harbour amplification of the MYCN oncogene [5].
Mutations activating the RAS/MAPK signalling pathway are frequent in high-risk and relapsed
neuroblastoma [6, 7], with relapsed neuroblastoma being almost always fatal. Most recently,
mutations in the p53/MDM2 or RAS/MAPK pathway in the presence of telomere mainte-
nance mechanisms were shown to define a subgroup of ultra-high risk neuroblastoma with a
5-year survival below 20%. Therefore, development of novel therapies for patients with high
risk or relapsed neuroblastoma is an urgent clinical need. Mutations of anaplastic lymphoma
kinase (ALK), present in 8% of all patients at diagnosis [8, 9], are the most common mutations
activating the RAS/MAPK pathway in neuroblastoma. In addition, mutations in PTPN11,
NF1, Ras and other RAS/MAPK pathway signalling elements occur in neuroblastoma [7, 10].

This makes RAS/MAPK pathway inhibition a promising treatment option for neuroblas-
toma, and ALK and MEK inhibitors are already being tested in early clinical trials [11]. How-
ever, tumour responses to targeted inhibitors were inconsistent, and early progression pointed
towards development of resistance, giving a strong incentive to understand mechanisms of pri-
mary and secondary resistance and how to overcome these mechanisms.

Resistance to targeted therapies of signalling pathways are often mediated by feedbacks that
re-wire or re-activate signalling. For example, resistance to PI3K/mTOR inhibition in breast
cancer is often mediated by feedbacks that lead to activation of JAK/STAT signalling [12]. Sim-
ilarly, in colon cancer, MAPK-directed therapy is counteracted by a negative feedback that
leads to hyper-sensitisation of the EGF receptor and ultimately reactivation of MAPK and
AKT signalling [13, 14]. Additionally, a very strong feedback from ERK to RAF leads to re-
activation of MAPK signalling upon MEK inhibition in many cancer types [15–17]. One
approach to overcome feedback-mediated resistance is by combinatorial therapy that co-tar-
gets the feedback [18].

We report here how a more quantitative understanding of feedback mechanisms might
help to optimise combinatorial treatment. We used a neuroblastoma cell line panel represent-
ing the class of very high-risk neuroblastoma, which we profiled for drug sensitivity, genomic
and transcriptomic alterations. We observed strong differences in the sensitivity to MEK inhi-
bition. To arrive at a mechanistic understanding of resistance to MEK inhibition, we generated
systematic perturbation data and quantified signalling using data-driven models. By this we
described qualitative and quantitative differences in feedback structures that might confer the
observed robustness to MEK inhibition. We then identified potential combinations capable of

sensitising highly resistant cell lines to MEK inhibition, and tested these combinations systematically.

## Results

### Drug sensitivity in a panel of very-high-risk neuroblastoma cell lines

We collected a panel of 9 neuroblastoma cell lines (CHP212, LAN6, NBEBC1, SKNAS, NGP, SKNSH, N206, KELLY and IMR32) and performed molecular profiling of these cells (RNA-sequencing and exome sequencing, see Fig 1A and S1 File). We noticed that all cell lines harbour a mutation in at least one of the RAS pathway genes with all cell lines having a mutation in either KRAS, NRAS, NF1, BRAF or ALK. One cell line (IMR32) had two mutations in the pathway: a mutation in KRAS and an atypical BRAF mutation. Most cell lines also have a mutation in one of the p53 pathway genes: ATRX, ATM, ATR, PRKDC, CDKN2A and TP53. Additionally, all express telomerase as seen by TERT expression, except for LAN6 which is known to have an alternative mechanism to lengthen the telomeres (ALT) [4]. We saw strong variability in the expression of MYCN, with 4 cell lines expressing low levels of MYCN, and 5 cell lines displaying high levels of MYCN. When considering mutations of individual genes, we found a strong heterogeneity within our panel, but overall the frequency of mutations in individual genes reflects that of high risk tumours [6]. Taken together, those data indicate that the chosen cell line panel can be seen as representative for the group of very-high risk neuroblastoma.

To further characterise the cell line panel, we measured drug sensitivity for 6 inhibitors that target components of the pathways shown to be affected by mutations (MAPK/PI3K/mTOR), using live cell imaging and computing growth rates from confluency measurements (Fig 1B). In this panel of cell lines, there was no notable difference in the sensitivity to the AKT inhibitor MK2206 or to the RAF/pan-tyrosine kinase-inhibitor Sorafenib. In contrast, pronounced variation in IC50 across the panel can be seen for mTORC1 inhibitor Rapamycin and MEK inhibitor AZD6244. When comparing to published drug sensitivity data, the IC50 for AZD6244 largely correlate with those derived for a different MEK inhibitor (binimetinib) [19]. All 6 NRAS wild type cell lines showed similar sensitivity to Rapamycin while the 3 NRAS mutant cell lines exhibited either strong resistance (SKNSH and SKNAS) or sensitivity (CHP212). This is only partly in agreement with previous literature that described CHP212 but also SKNAS as sensitive to sub-nanomolar concentrations of Everolimus, a Rapamycin analog [20]. AZD6244 is the drug with the most variable drug response, with a subset of 6 cell lines cell lines being very resistant to AZD6244 (IC50 $>10\mu$M, Fig 1C and S1 Fig) and another subset of 3 cell lines showing extreme sensitivity (IC50 $\approx$ 10–100 nM). When correlating inhibitor sensitivity with mutations, we found no notable correlation for AZD6244 and Rapamycin (S2 Fig). Drug sensitivities also did not correlate significantly with selected expression data (adjusted p>0.93 for the 1000 most variable genes and adjusted p>0.94 for GO signal transduction genes, S3 Fig). Also a PCA analysis could not separate cells according to MEKi sensitivity for those two expression groups (S4 and S5 Figs). For instance, previous reports showed that NF1 expression is linked to sensitivity to MEK inhibitors [19], however we only found a weak and non-significant correlation with AZD6244 sensitivity ($R^2 = 0.34$, $p = 0.10$, S6 Fig). Taken together, this data establishes that this cell line panel represents a heterogeneous group of very high risk neuroblastoma that differ in drug sensitivity, most prominently against MEK inhibitors. Furthermore, it suggests that the difference cannot be explained by single mutations or expression of marker genes alone.

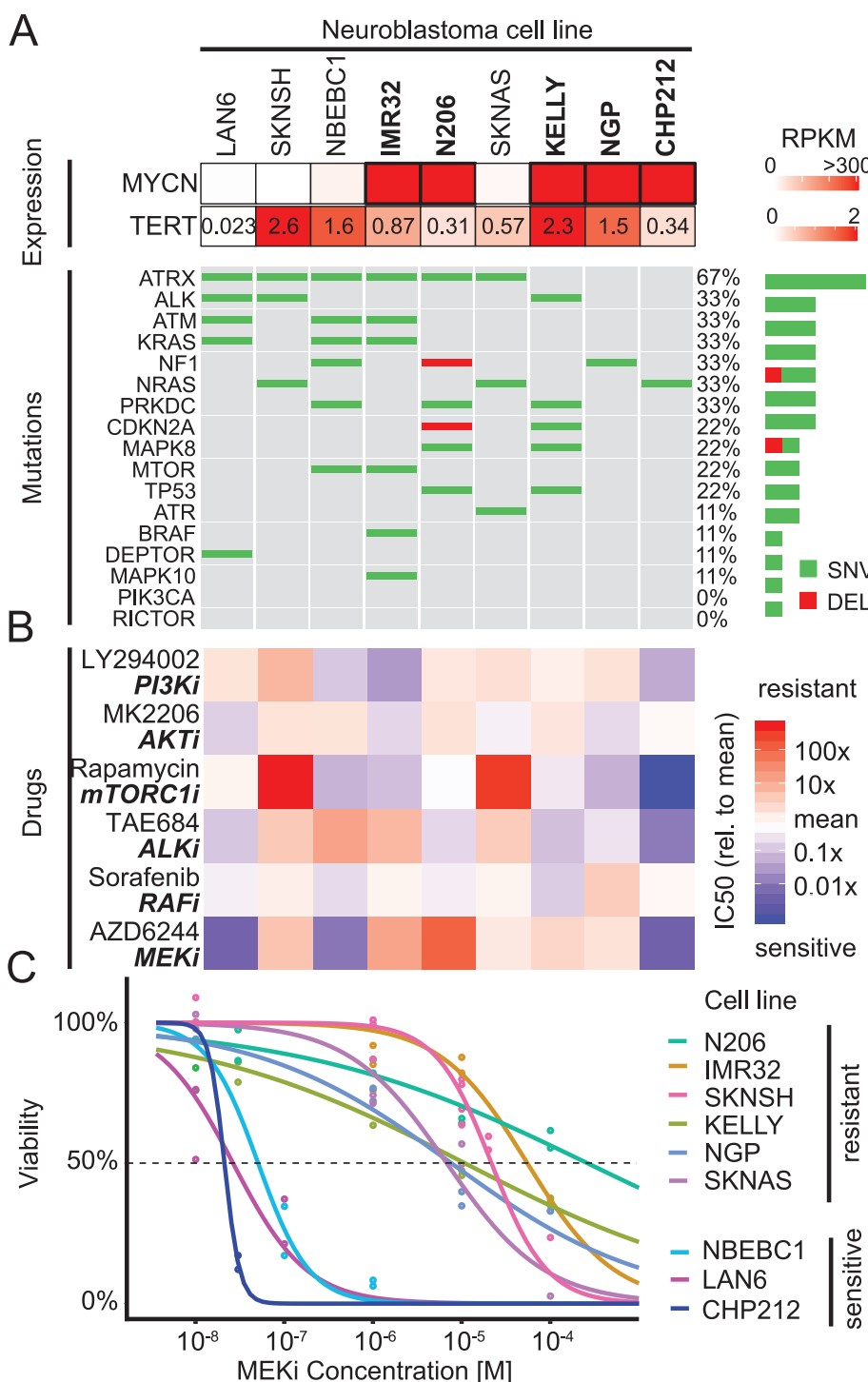

**Fig 1. Mutations are insufficient to explain sensitivity variations to RAS/PI3K drugs in neuroblastoma cell line panel.** A: Oncoprint of 9 neuroblastoma cell lines for RAS/p53/PI3K related genes along with MYCN and TERT mRNA expression. Bold font indicates MYCN-amplified cell lines. B: Relative IC50 of the same 9 neuroblastoma cell lines as in A for drugs targeting the PI3K and MAPK pathways (n = 2). C: Viability concentration curves for the MEK inhibitor AZD6244 on the neuroblastoma cell line panel along with the calculated IC50 (intersection with dotted line). Points represent measurements (n = 2).

## Perturbation-response data unveils heterogeneity in signalling

To get insights into the underlying mechanisms of resistance to the MEK inhibitor AZD6244, we selected 6 neuroblastoma cells lines that represented the spectrum of sensitivity to MEK inhibition (sensitive: CHP212, LAN6; resistant: SKNAS, SKNSH, KELLY and IMR32). Using these cell lines, we performed perturbation experiments, in which we stimulated the cells by growth factors for 30 minutes, and additionally inhibited specific pathways for 90 minutes (Fig 2A). After perturbation, we then monitored pathway activity by measuring phospho-proteins.

We designed the experiments such that they probe the AKT/mTOR and MAPK signalling pathways (Fig 2B). Specifically, we selected ligands that might activate those pathways based on the expression of growth factor receptors in the cell lines. As expression of receptors was heterogeneous (S7 and S8 Figs), we chose a set of growth factors such that each cell line had robust expression of receptors for at least two provided ligands. Inhibitors were chosen such that they block key steps of the pathway. The position of perturbations and readouts in the signalling network is shown in Fig 2B. We perturbed the 6 cell lines with 4 ligands (PDGF, EGF, IGF1 and NGF, shown in blue) and 7 inhibitors (GS4997 (ASK1i), MK2206 (AKTi), Rapamycin (mTORC1i), AZD6244/Selumetinib (MEKi), Sorafenib (RAFi), TAE684 (ALKi) and GDC0941 (PI3Ki), shown in red) alone or in combinations. Subsequently, we measured 6 phosphoproteins (MEK, ERK, AKT, S6K, p38 and cJUN, yellow background) for each perturbation using a sandwich ELISA where a first bead-bound antibody captures the protein and a second recognises the phosphosite of interest. All experiments were performed in two biological replicates.

Overall, the perturbation experiments yielded 240 data points per cell line, which are visualised in a heatmap in Fig 2C. Inspection of the heatmap shows that the perturbation-response data has similar patterns in different cell lines, but there are also clear differences. For instance, inhibition of mTOR leads to down-regulation of phospho-S6K across all cell lines, but inhibition of AKT and PI3K has diverging effects on S6K. Similarly, application of MEKi leads to an increase of phospho-MEK across all cell lines, but ALK inhibition had varying effects in different cell lines.

To get further insights into this high-dimensional data set, we performed principal component analysis (PCA) on the perturbation data (Fig 2D top and S9 Fig). The PCA highlights 3 groups of cell lines. The first component (42% of variance) separates the cell lines according to the effect of Sorafenib and TAE684 on AKT and S6K. The second component (26%) separates IMR32 and KELLY based mainly on the MEK response to MEK inhibition. The third component (18%) contains the effects of IGF1, GS4997 and Rapamycin on AKT and S6K and mainly separates KELLY and IMR32 (S10 Fig).

When we applied hierarchical clustering on the cell line panel, SKNSH was clustered separately, suggesting that it has a very atypical response to the perturbations, with a generally very high response to all ligands, and an especially strong response to PDGF (Fig 2D bottom). This atypical status of SKNSH is also present in the mRNA expression, with a PCA on the most variables genes or on the genes in the GO term "signal transduction" separating it from the other cell lines. Interestingly, CHP212 also separated from the other cell lines in a PCA based on gene expression data, but not when considering the response to the perturbations. When grouping cells by MEK inhibitor sensitivity, we noticed that simple multivariate analysis by PCA does not separate cells into groups that correspond to sensitive or resistant cells (Fig 2D top and S9 Fig), and also hierarchical clustering does not separate sensitive from resistance cell lines (Fig 2D bottom).

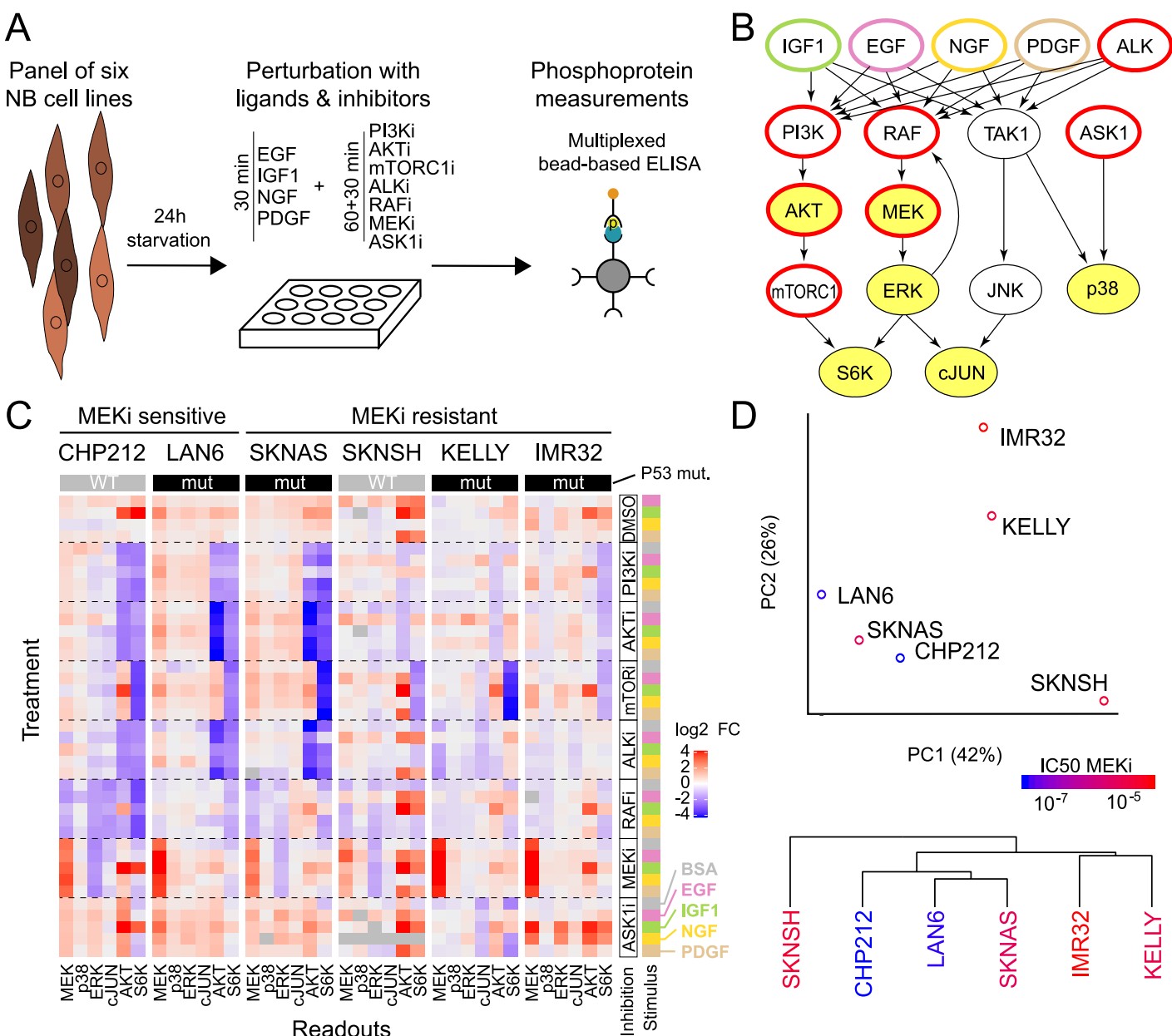

**Fig 2. Neuroblastoma cell lines show heterogeneous responses to signalling perturbations.** A: Outline of the perturbation experiments. A panel of cell lines was treated with growth factors and small molecule inhibitors, and the resulting effect on selected phosphoproteins was measured using multiplexed bead-based ELISAs. B: Graphical representation of the perturbation scheme on a literature signalling network. Blue and red contour highlights ligand stimulation and kinase inhibition, respectively; yellow filling shows measured phosphoproteins. C: Perturbation data obtained from applying all combinations of 4 ligands or BSA control and 7 inhibitors or DMSO control to 6 neuroblastoma cell lines. Each measurement is normalised by the BSA+DMSO control of the corresponding cell line and represents at least 2 biological replicates. Readouts are phospho-proteins p-MEK1$^{S217/S221}$, p-p38$^{T180/Y182}$, p-ERK1$^{T202/Y204}$, p-cJUN$^{S63}$, p-AKT$^{S473}$ and p-S6K$^{T389}$. D: Global non-mechanistic analysis of the perturbation data presented in C: TOP first two components of a principal component analysis and BOTTOM hierarchical clustering. Colour scale corresponds to the IC50 for AZD6244 treatment (see also Fig 1C).

## Signalling models highlight differential feedback regulation of MEK

To get further, more mechanistic, insights into potential resistance mechanisms, we used the perturbation data to parameterise signalling models. We applied our previously developed method that has been derived from Modular Response Analysis (MRA, implemented as R

package STASNet, [21]) to fit signalling network models to each cell line. This modelling procedure requires a literature network and the perturbation data as input, and then estimates response coefficients corresponding to link strengths using a maximum likelihood estimate (see Fig 3A, first step). By using the statistical framework of the likelihood ratio test, the modelling procedure then allows to test if any extension of the literature network is required to describe the data (see Fig 3A, second step). To compare parameters between cell lines, it is essential to harmonise parameters between all cells that can practically not be identified alone, i.e. parameters for inhibitors (see Fig 3A, third step). This finally yields a parameter map that allows to compare signalling strength between cell lines (see Fig 3A, final step).

When starting with a canonical literature network (see Materials and methods), we obtained reasonable fits for 4 of the 6 cell lines, as judged by the sum of weighted squared residuals that is in the order of number of data points (Fig 3B, red bars), and the normal distribution of residuals (S11 Fig). When we systematically tested if extensions of the network improve the fit using a likelihood ratio test, we found that significant improvements were still possible for most cell lines. We therefore performed successive rounds of extensions for each cell line independently (Fig 3A and S3 File). While SKNSH required no extension of the literature network, CHP212, LAN6, SKNAS required two or three extensions. KELLY and IMR32, the two cell lines that initially had the poorest fit, required four extensions (Fig 3C). After the extension the sum of weighted squared residuals was in the order of the number of data points for all cell lines except KELLY (Fig 3B green bar). The high residuals still exhibited by KELLY could be narrowed down to uncertainties in individual data points (see S3 File). Two network extensions (ASK1→MEK and p38→S6K) were significant in at least 3 cell lines and correspond to an effect of the ASK1 inhibitor GS4997 on the MEK/ERK MAPK pathway and S6K. Both links are negative which suggests an antagonism between the p38 MAPK and the MEK/ERK MAPK pathways in neuroblastoma cell lines. This negative crosstalk from p38 to MEK/ERK has also been described in other cell systems, e.g. after p38 knockdown in HeLa cells [22].

All extended models had similar, but different, parameters for the inhibitor strength. However, there is a strong interdependence of the inhibitor strength and link strength downstream of the inhibitor which render comparison between those link strengths in different cells difficult (see S3 File). As all cell lines received the same inhibitor concentration we therefore harmonised the inhibitor parameters by fixing them to the mean value between all models (Fig 3A, fixed inhibitor parameters). The resulting harmonised models maintained a good agreement with the data (Fig 3B, blue bars) and were used for inter-model comparisons (Fig 3D and 3E).

When inspecting the parameters for ligand-induced pathway activation, we noticed that they reflected a strong heterogeneity in ligand response between the cell lines. Reassuringly, they matched the expression of the corresponding receptors in many cases (Fig 3D and S12 Fig). The parameters for pathways downstream of NGF correlated mostly with NTRK1 expression and not with NGFR expression, which might indicate that NGF signalling is mediated mostly via NTRK1 in those cell lines. The parameters for IGF-induced signals correlated with IGF1R or IGF2R for MEK and AKT, respectively, indicating that both receptors mediate IGF1 signalling independently. Interestingly, the parameters for the pathway from EGF to MEK did not correlate with EGFR expression, but they do for EGF to AKT, which might suggest that differences in adaptor protein expression shape routing into downstream signalling in the various cell lines. Indeed, the expressions of GAB2 and SRC are very different between the cell lines and could explain that IMR32 and LAN6 are activated by EGF as strongly as SKNAS and SKNSH despite their lower EGFR expression (Fig 2C and S6 Fig). Another potential cause for the attenuated activation of MEK/ERK is that in NRAS mutant cell lines (CHP212, SKNAS and SKNSH), MEK/ERK activity is less inducible by receptors, as also parameter values of the

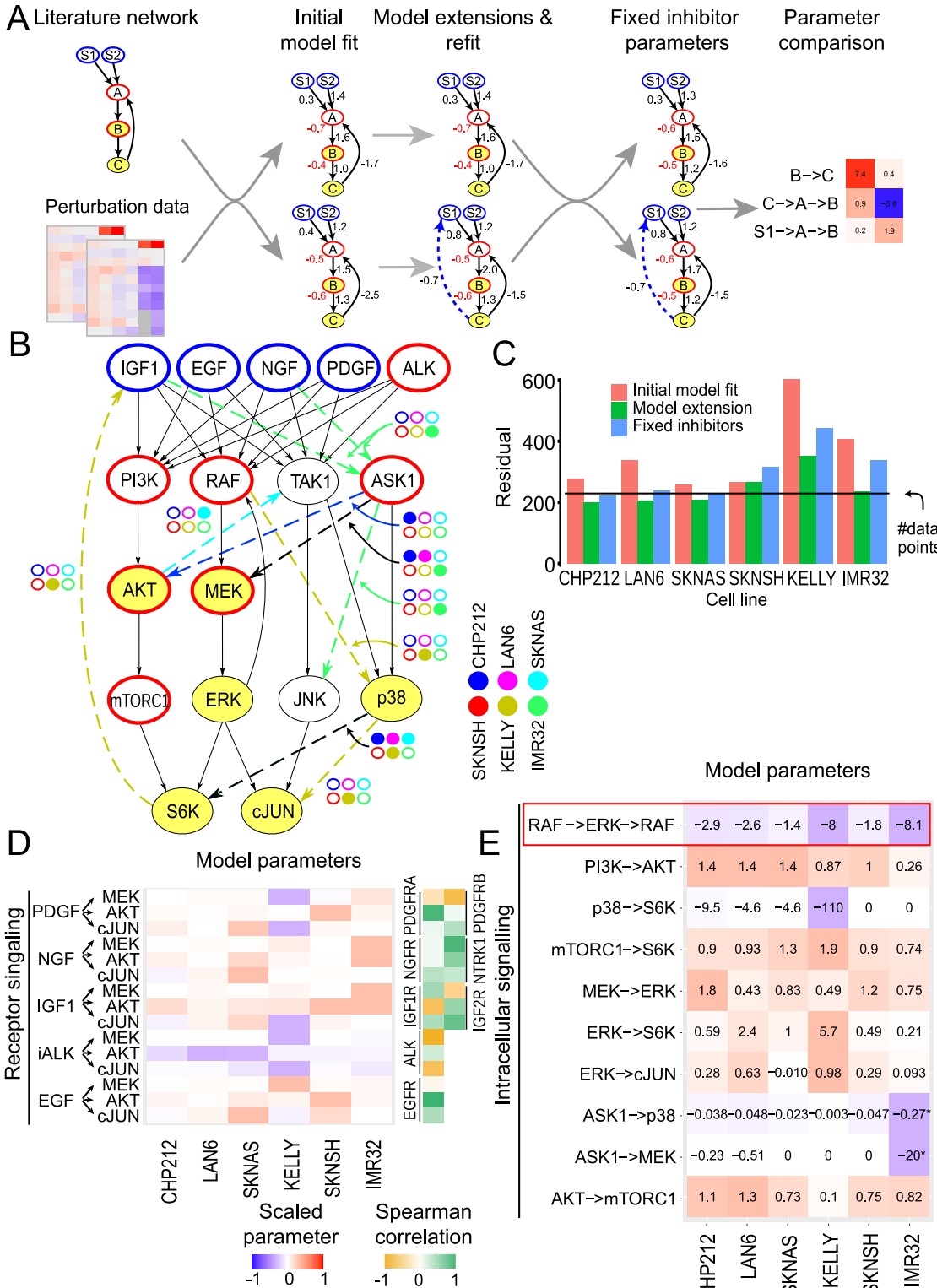

**Fig 3. Receptor expression and topology variations explain the heterogeneity in perturbation response.** A: Starting from a literature-derived network, a model was fitted for each cell line (Initial model fit) and extended following suggestions from the model (Model extensions and refit). Those models with different network structures were then harmonised by fixing the inhibition parameters to a consensus value (Fixed inhibitor parameters) to make the parameters directly comparable (Parameter comparison). B: Model residuals before and after model extension and harmonisation. The black line represents the number of data points, which

is equal to the expected mean of the error if the model explains all the data. C: Cell-line-specific network extensions (dashed arrows) relative to the literature network. Colour of the extended link was matched to cell line colour if required in only one cell line model and black otherwise. D: Model paths from the receptors to the first measured downstream node and correlation with the corresponding receptor expression. The colours correspond to the value of the path scaled by the maximum absolute value of that path between all cell lines. E: Model paths between non-receptor perturbed nodes and measured nodes for routes present in at least 2 cell lines. Colour scale is the same as in D. Cells are ordered from left to right from most sensitive to most resistant to the MEK inhibitor AZD6244. Due to the absence of ASK1 basal activity in IMR32, ASK1->p38 and ASK1->MEK represent in this cell line NGF->ASK1->p38 and NGF->ASK1->MEK, respectively.

routes from PDGF, EGF, NGF and IGF into MAPK signalling are lower in those cell lines. Conversely, these cell line models display a slightly more inducible PI3K pathway. This observation is in agreement with a recent comparative study of G12V-mutated RAS isoforms in colorectal SW48 cells, where the NRAS-mutated cell line showed a weaker coupling of receptors to MEK and a stronger coupling to PI3K than in the parental cell line [23]. This would suggest that an activation of the MEK/ERK pathway is relayed predominantly by NRAS while the PI3K pathway activation is mediated by other proteins [24]. Taken together, this shows that the wiring and routing of ligand induced signalling in these cell lines is varying and is mostly explainable by the expression of the corresponding receptor and RAS mutation status.

In contrast to the receptor-associated parameters, the strength of intra-cellular kinase paths are less variable, and most paths are comparable between cell lines (Fig 3E). The most prominent exception is the negative feedback in MAPK signalling from ERK to RAF. When compared to the other cell lines, this feedback appears to be 3 to 4 times stronger in KELLY and IMR32, which are two cell lines that are highly resistant to AZD6244. A strong RAF-mediated feedback is a known resistance mechanism against MEK inhibitors [15–17], where relieve of inhibition of upstream components post inhibition can partially reactivate signalling. This suggests that AZD6244 resistance could be mediated by a differential regulation of this feedback.

Apart from the RAF-mediated feedback, MAPK signalling is also controlled by receptor-mediated feedbacks. In the KELLY cell line, our modelling procedure extended the model by a negative feedback from S6K to IGFR that could then explain the strong accumulation of pMEK by IGF following AZD6244 treatment (Fig 3C and S3 File). Receptor-mediated feedbacks are also known to mediate resistance, notably to MAPK inhibitions [13, 18, 25–27], by reactivating this pathway and other parallel pathways.

In summary, the signalling parameters derived from the perturbation data by our models show that cell lines diverge in receptor expression and feedback regulation, with strong multi-layered feedbacks for some of the resistant cell lines.

## Differential quantitative wiring of resistant cell lines

A hallmark of negative feedbacks is that they lead to re-activation of the pathway after pathway inhibition. In agreement with this, we observe an increase of phosphorylated MEK upon MEKi treatment (AZD6244) that is more pronounced in the cell lines IMR32 and KELLY compared to the other cell lines modelled, including the most sensitive cell lines CHP212 and LAN6 (Fig 4A and S13 Fig). We also tested the most resistant cell line in our panel, N206, which also showed a strong feedback response (Fig 4A). To more precisely dissect the feedback wiring, we generated additional focused perturbation data for those cells with high feedback (KELLY, IMR32 and N206) to MEK inhibition. We stimulated cells with different growth factors (IGF and NGF or EGF), and blocked MAPK signalling with MEK and RAF inhibitors, and subsequently monitored six phosphoproteins (Fig 4B). Subsequently, we used this data to parameterise a focused MRA model that additionally either contained or did not contain the only receptor-mediated feedback found in the first modelling round from S6K→IGF1 (Figs

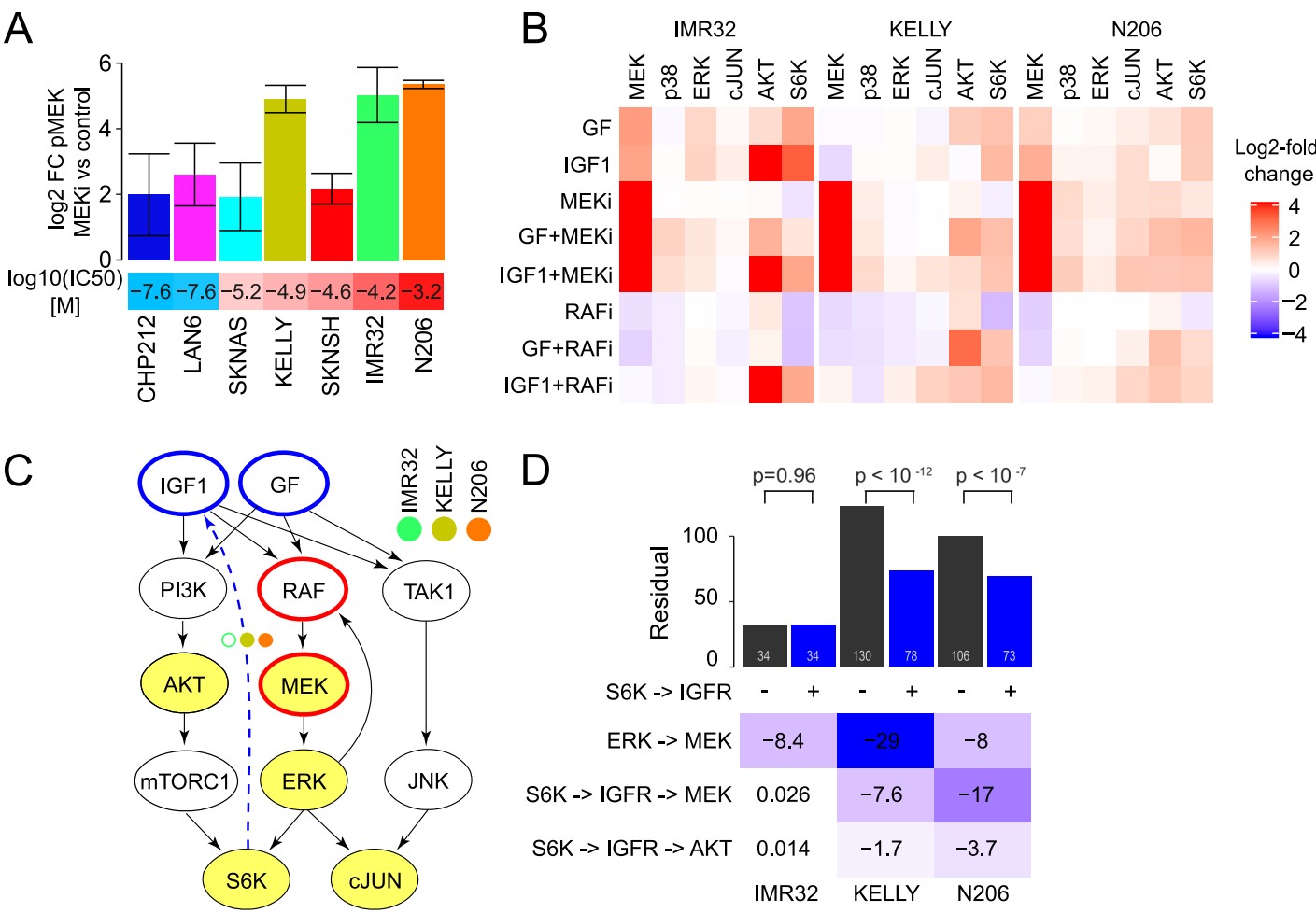

**Fig 4. AZD6244 resistant cell lines have strong feedback control of MAPK signalling.** A: Mean pMEK log2-fold change relative to control after AZD6244 treatment in 7 neuroblastoma cell lines measured with bead-based ELISAs. Error bars represent 95% confidence interval. B: Measurement of 6 phosphoproteins (columns) after perturbation of N206, IMR32 and KELLY by either EGF (KELLY, N206) or NGF (IMR32) (together referred to as GF), IGF1, or control BSA in combination with Sorafenib (RAFi), AZD6244 (MEKi) or control DMSO. Values are expressed in log2-fold change to BSA+DMSO control. C: Starting model and S6K→IGF1 receptor extension for the high pMEK responder cell lines. D: (top panel) Model residuals for N206, IMR32 and KELLY models with (black) or without (blue) an S6K→IGF1 receptor feedback link and corresponding p-value($\chi^2$ test with df = 1). (bottom panel) Parameter values of the high pMEK responder models including the S6K→IGF1 receptor link.

3C and 4A). Inclusion of the IGF receptor-mediated feedback led to a significantly better fit of the data for N206 and KELLY ($\chi^2$ p<0.05), but did not improve the IMR32 model (Fig 4C and 4D). Interestingly, the S6K→IGF1→RAF→MEK feedback is stronger in the N206 models, but the pathway-intrinsic feedback (ERK→RAF→MEK) is stronger in KELLY (Fig 4D). This highlights that all these cells display negative feedback regulation, but the strengths of the two layers of feedbacks are different between cell lines.

## Parallel inhibition of MEK and IGFR leads to synergistic effects on the phosphoproteome

To gain a more systematic understanding of the effect of MEK and IGFR inhibition on the signalling states of the cells, we generated deep (phospho-)proteomics profiles using tandem mass-tag (TMT) based mass spectrometry [28, 29]. We reasoned that inhibition of IGFR in

combination with MEK should have a synergistic effect in N206 compared to IMR32. We measured the phospho- and total protein levels in IMR32 and N206 cells after 4h treatment with MEK and/or IGFR inhibitors and control cells. Although a similar number of phosphosites were dis-regulated in both cell lines (448 in IMR32, 615 in N206, FDR < 0.05), there was little overlap in the phospho-peptides differentially regulated between the two cell lines (Fig 5A), and this overlap was mostly limited to phospho-peptides affected by MEK inhibition (S14 Fig). In IMR32, IGFR inhibition had little effect, while the presence of MEK inhibition strongly affected the phosphoproteome (Fig 5B left). Moreover the effect of the combination of MEK and IGFR inhibitors was dominated by the effect of the MEK inhibition, with about two thirds of the differential phosphopeptides (96/149) being also regulated by MEK inhibitor alone. Accordingly, differentially phosphorylated peptides in IMR32 are enriched in MAPK targets (S15 Fig). In contrast, both MEK as well as IGFR inhibition induce strong alterations in the phosphoproteome in N206 (S14 Fig), affecting both mTOR and MAPK signalling targets (S15 Fig), and the combination exhibits a synergistic effect (Fig 5B right). Overall, 24 differentially phosphorylated sites in N206 show synergistic regulation, as defined by a significant deviation of the combination from the sum of the individual treatment effects. Of these, 17 phosphosites were synergistically down-regulated, and 7 sites showed up-regulation. Moreover, 10 of those synergistically downregulated phosphosites are known or putative targets of the PI3K/AKT signalling axis. This suggests that MEK/ERK signalling influences AKT signalling in a IGFR dependend way. In contrast, only two sites showed synergy in IMR32 (Fig 5C). Among the synergistically downregulated phospho-sites in N206 was S425 of the Eukaryotic translation initiation factor 4B (EIF4B), a protein involved in regulation of translation and a known nexus between AKT and MAPK signalling [30]. We performed a kinase substrate enrichment analysis [31] to explore how the signalling networks were affected by the inhibitions (Fig 5D). For IMR32 cells, this analysis showed a decreased phosphorylation of MEK and JAK targets and an increased phosphorylation of ARAF and BRAF targets in response to MEK inhibition. Interestingly, in combination with IGFR inhibition the RAF activation is partially reversed whereas other kinase targets seem rather unaffected. Overall this indicates a feedback activation of RAF that does not totally compensate the loss of MEK activity. In N206 cells, the response to MEK inhibition and the attenuation of the activation of RAF targets following double inhibitor treatment is similar to the response in IMR32. However, in IMR32 cells IGFR inhibitor treatment had little impact on the kinome whereas a massive down-regulation of targets of a range of kinases occurred in N206 cells, covering the PI3K/AKT/mTOR pathway (SGK1–3,AKT1,p70S6K), MAPK pathway (p90RSK) and many members of the Protein Kinase C Family. This suggests a central role of IGFR signalling on central growth and survival pathways.

When we investigated the phosphorylation of components of the MAPK pathway more closely, we found many RAF negative feedback/crosstalk sites to be down-regulated after MEK inhibition (BRAF: T401, S750, T753; RAF1: S29, S642, S259) in both cell lines (Fig 5E). MEK1 S222/S226 phosphorylation is increased and pERK S204 decreased in both cell lines after MEK inhibition, in line with corresponding measurements using bead-based ELISAs. Among those down-regulated phosphosites that were only significant in the combination in N206 we detected many MYCN-phosphosites, notably MYCN S62, which is regulated by MAPK via CDK1 [32]. Interestingly, this loss of S62 phosphorylated MYCN is associated with reduced MYCN levels (Fig 5F) despite the association of MYCN S62 with increased MYCN degradation [33]. The decreased detection of MYCN S62 might be a consequence of the loss of total MYCN protein but is likely not causing this loss itself. This downregulation was observed in IMR32 and N206 cells upon single inhibition (IGFRi for N206 and MEKi for both cell lines), but only in N206 cells an even stronger downregulation could be observed upon double

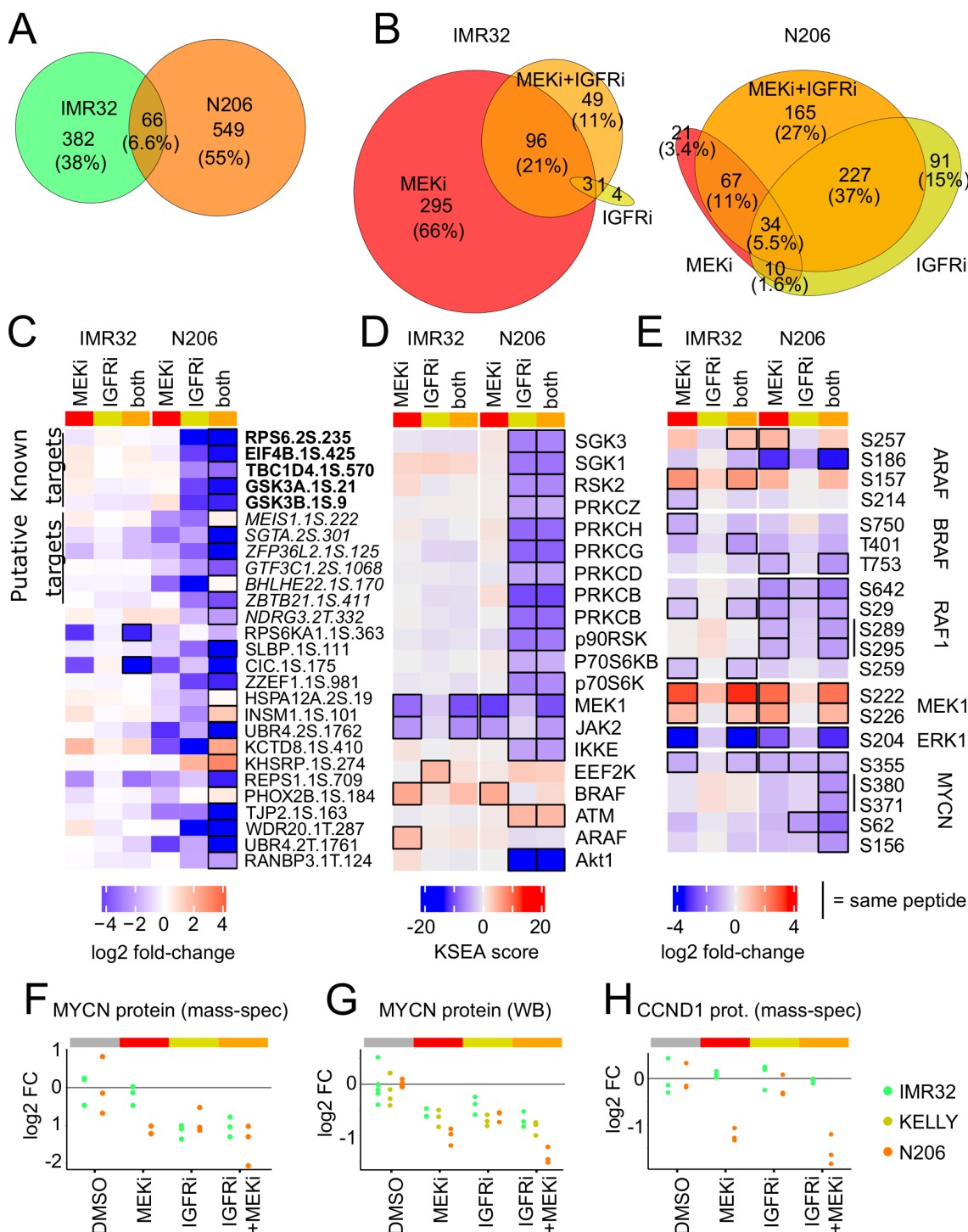

**Fig 5. Phosphoproteomics analysis reveals important variations in the response to combination treatment.** A-B: Venn diagrams showing the overlap in differentially regulated phosphosites A: between IMR32 and N206 or B: between treatments for each cell line. C: Phosphopeptides synergistically altered by MEKi+IGFRi combination (black outline) when compared to the sum of individual inhibitor treatments. AKT, mTOR or P70S6K *bona fide* targets (bold font) and putative targets (italic font; top 5 predicted kinases by PhosphoNET Kinase Predictor www.phosphonet.ca) are indicated. D: Kinase substrate enrichment score using PhosphoSitePlus annotations. E: Log-fold change to DMSO for RAF/MAPK and MYCN phosphopeptides. C-E: Black outline highlights significant changes in activity (limma moderated t-test, FDR<5%) F-H: Relative levels compared to control of the total proteins levels for MYCN (F) and CCND1 (H) measured with mass spectrometry and MYCN measured with Western blot (G).

inhibition (Fig 5F). We confirmed these effects in Western blots for IMR32 and N206 cells (Fig 5G), and also found downregulation of MYCN upon IGFRi as well as MEKi treatment but no synergetic decrease after the combination treatment (Fig 5G). Another interesting protein that is regulated synergistically in N206 is Cyclin D1 (Fig 5H), a protein that is involved in cell cycle progression and whose loss likely mediates MYCN loss. It should be noted that only 5 proteins (PHGDH, DERL1, AMPD3, ARHGEF16 and CCND1) were found differentially affected with an FDR < 10%, highlighting that on this time scale phospho-protein changes dominated.

Taken together, the proteomics data is coherent with the model that MAPK signalling in N206 is controlled by a dual feedback structure involving RAF and IGFR, whereas it is mainly controlled by a RAF-mediated feedback in IMR32. It furthermore supports the notion that treatment with MEK and IGFR inhibitors would show synergy in N206.

## Vertical inhibition can break feedback-mediated resistance

Feedback regulation is often a central aspect for drug resistance that could be overcome by a vertical inhibition strategy, where an inhibition of an upstream node prevents pathway reactivation. Based on our models, we tested if the additional application of an inhibitor targeting the feedback nodes (RAF and IGFR) would sensitise resistant cells toward MEK inhibition (Fig 6A). We quantified growth reduction after inhibiting IMR32, KELLY and N206 with different dose combinations of inhibitors against MEK (AZD6244), IGFR (AEW541) and RAF (LY3009120) (Fig 6B). In agreement with our model predictions of strong IGFR-mediated feedback in N206 (Fig 4D), there was a strong synergistic effect as evaluated by the Bliss score [34] of the combination of MEK and IGFR inhibitions on growth in N206 but little in KELLY or IMR32 (Fig 6C and see S16 Fig for Loewe score).

When trying to overcome the model-derived strong ERK-RAF feedback found in all three cell lines with a combination of MEK and RAF inhibition we only found a synergistic effect for two of the three cell lines (N206 and KELLY), whereas IMR32 remained resistant and no synergy could be detected. We hypothesised that this observed resistance in IMR32 might be either because the vertical inhibition by MEKi and RAFi was molecularly not effective or that IMR32 might no longer depend on ERK signalling for survival and growth. To distinguish the former from the latter we decided to compare model simulation and measurements for perturbation effects of selected inhibitor combinations on pMEK and pERK in IMR32 and KELLY cells.

Based on the model simulations, in both cell lines the vertical inhibition of MEK + RAF inhibitor was predicted to suppress MAPK signalling much stronger than MEK inhibitor alone or in combination with an ERK inhibitor. Moreover, the suppressive effect was predicted to be even more profound in IMR32 than in KELLY (Fig 6D left). We then measured the effect on pMEK and pERK of MEK inhibitor alone and in combination with the RAF inhibitor LY3009120 or ERK inhibitor SCH772984 (Fig 6D right). The measurements qualitatively supported the model simulations showing that RAF inhibitor suppressed MEK feedback activation by AZD6244, and that this suppression is stronger in IMR32. Addition of the ERK inhibitor neither suppressed this feedback activation nor could it decrease ERK phosphorylation more than RAF inhibition, as also predicted by the model. This suggests that in agreement with the model simulations the combination of RAFi and MEKi is most effective in IMR32 to effectively suppress ERK activation and feedback-mediated re-activation. However, since the growth is least affected by this combination IMR32 seems not to depend on ERK activity.

As both KELLY and N206 have strong multi-layered feedbacks (Fig 4D), we also tried triple combinations of IGFRi, RAFi and MEKi. We observed that only in KELLY, triple inhibitor

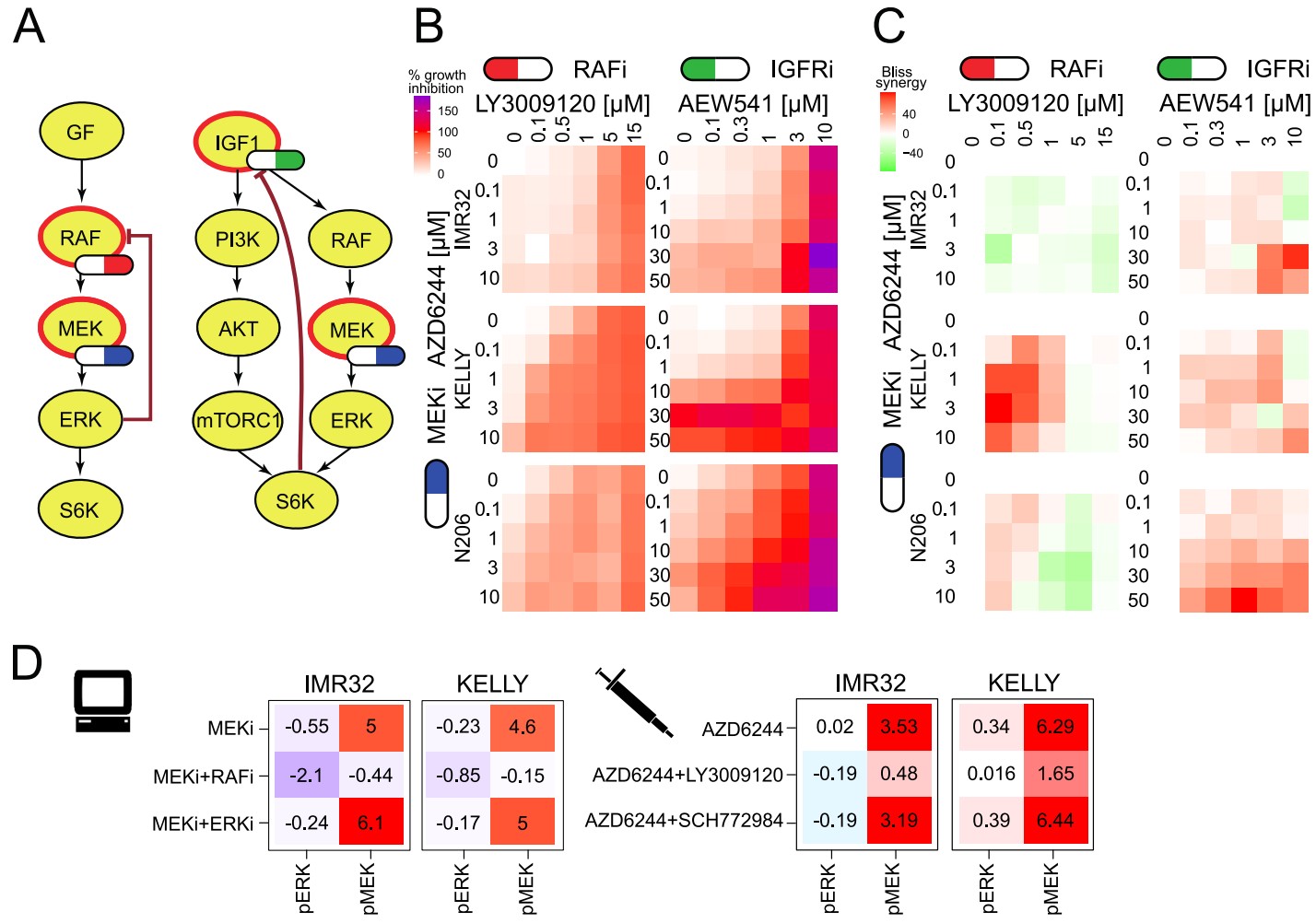

**Fig 6. AZD6244 resistant cell lines can be sensitised with combined inhibition with the IGFR inhibitor AEW541 or the RAF inhibitor LY3009120.** A: Model-inferred targeting strategy of dual inhibition. B: Growth inhibition measurements for various combinations of the MEK inhibitor AZD6244 with the RAF inhibitor LY3009120 or the IGFR inhibitor AEW541. Values over 100 indicate cell death. n = 2. C: Bliss synergy corresponding to the measurements in B. D: LEFT: Model predictions of pERK and pMEK activity for MEK inhibition alone and in combination with inhibition of upstream kinase RAF or downstream kinase ERK for KELLY and IMR32. Values are log-fold changes to IGF1 condition with inhibitor strength set to -1. D: RIGHT: pERK and pMEK plex measurements in KELLY and IMR32 after 90min treatment of the MEK inhibitor AZD6244 in combination with either DMSO, SCH772984 (ERKi, $10\mu M$) or LY3009120 (RAFi, $5\mu M$) in cells grown with 10% FCS. Values are log-fold change to FCS medium condition.

treatment seems to have an additional benefit compared to the best combination of two inhibitors. (S17 Fig and S2 File).

## Discussion

Neuroblastoma is a complex disease with distinct subtypes that display radically different outcomes, ranging from spontaneous regression in low-risk groups to only 50% survival of patients in the high risk neuroblastoma group. Mutations in RAS/MAPK signalling are a hallmark of high risk neuroblastoma, and also define a subgroup of patients with ultra-high-risk neuroblastoma and an even worse survival. Therefore targeted treatment might be a valid strategy to treat those patients. However, response to MEK inhibitors are very variable, and it is thus important to understand mechanisms of resistance and how to circumvent these.

In this work, we explored how a more quantitative understanding of signalling can be used to design combinatorial treatments to counteract drug resistance. We used a panel of deeply profiled cell lines representing high risk neuroblastoma and showed that the response to MEK inhibitors is variable, with some cell lines responding at low doses in the nM range, whereas others are highly resistant. By using signalling perturbation-response data, we characterised the signalling network surrounding MAPK. Analysis of that perturbation data with the modelling framework of modular response analysis unveiled that MAPK signalling is controlled by a multi-layered feedback with variable strength. A central finding was that MEK-inhibitor sensitive cells are controlled by low feedbacks within the MAPK cascade, whereas a subset of resistant cell lines shows strong multi-layered feedbacks that may be causal for resistance. Simulation of cell-line specific models suggested that different combinations of inhibitors can be used to overcome resistance, and experiments could confirm these predictions in two out of three cell lines.

Our work highlights that systematic perturbation data are a powerful source to probe intracellular signalling pathways. The connectivity of signalling pathways implies that minor quantitative alterations of the network can lead to many changes in response, not all of which alter the phenotype. In this work, we saw that multivariate analysis of the perturbation data alone was not fruitful to separate cell lines with respect to their drug sensitivity. In contrast, integration of data by models highlighted that variations of only a few links is enough to explain the differences between those cell lines. Modelling was therefore key to integrate the data and to unveil feedback loops as potential sources of resistance.

In our work we used a maximum likelihood version of MRA, but there are multiple other methods that might be suited to reconstruct semi-quantitative signalling networks from perturbation data. [35] proposed a bayesian variant which overcomes the linearity assumption of MRA using chemical kinetics to guide the inference and fuzzy-logic models such as used by [36] also show good performance to reconstruct network topology from signalling data. However getting quantitative values for the interactions between components of a signalling network from a small set of perturbations requires MRA variants [21, 37] or necessitates time-resolved perturbation data which limits the number of perturbations that can be studied simultaneously [38]. While boolean models are very good strategies to model large signalling networks and complex synergies [39], they would be unable to capture quantitative differences in feedback regulation, which are the key resistance mechanisms uncovered in this work.

Drug resistance to targeted therapies have been attributed to negative feedback loops in multiple tumours. Most importantly, sensitivity to MEK inhibitors is strongly influenced by a pathway-intrinsic feedback, where ERK phosphorylates RAF at multiple sites [15–17]. This feedback has been shown to be very strong in epithelial cells leading to pathway robustness [16], which can be overcome by vertical inhibition of RAF [17]. Another mode of feedback regulation is the inhibition of receptors by pathways. An example is the inhibitory regulation of EGFR by the MAPK pathway [13, 14]. When inhibiting MAPK signalling by MEK or RAF inhibitors, this feedback leads to hyper-sensitisation of EGFR, which in turn reactivates MAPK signalling and additionally activates other downstream pathways such as PI3K/AKT signalling. Also in this case vertical inhibition can help to overcome this mode of resistance, by co-targeting the MAPK pathway and the upstream receptor.

Our modelling analysis suggested that some neuroblastoma cell lines possess two major layers of feedback in MAPK signalling. One of these feedbacks is pathway-intrinsic (from ERK to RAF) and one is a feedback to the IGF receptor. Interestingly, different cell lines show different relative strength of feedbacks from ERK to RAF and IGFR, and simulations show that those require different strategies for vertical inhibition. For the cell line KELLY, modelling unveiled an extremely strong negative feedback from ERK to RAF. This suggests that a combination of

MEK and RAF inhibitor will be more potent than a combination of MEK and IGFR inhibitor. In contrast, in the cell line N206, both feedbacks have similar strength, suggesting that both combinations might be potent. In line with these predictions, experiments showed that in KELLY indeed the combination of MEK and RAF inhibitors is much more potent to reduce growth compared to the combination of MEK and IGFR. In contrast, in N206 both combinations reduce growth.

Our phospho-proteomics analysis shows that the combination of MEK and IFGR also has different effects in the two cell lines: Whereas it shows clearly synergistic effects of the combination in N206, there is no sign of synergy in IMR32. By aggregating the phosphoproteome to kinase activities using kinase enrichment scores, one can also get insights into the re-wiring of signalling after perturbation. In our case, it clearly shows how the re-activation of RAF after MEK inhibition is inhibited by the treatment with IGFR inhibitors, and IGFR and MEK inhibitors synergize in reducing AKT activity in N206. The phosphoproteome also showed that the dual treatment of IGFR and MEK manifests itself in synergistic downregulation of important proteins that are regulated by convergent signalling of MEK and AKT, such as MYCN and EIF4B.

Interestingly, a third resistant cell line, IMR32, showed no response in growth to MEK inhibitor in vertical combination with either RAF and/or IGFR inhibitor on growth, even though it's cellular ERK signalling was strongly responsive. This highlights that cancer cells might lose ERK-mediated cell cycle control, suggesting that coupling of cellular phenotype to signalling pathways is not necessarily strict [40, 41]. To more directly model changes on cellular phenotypes such as growth or viability, models of signalling would need to be connected to phenotypic readouts [42]. In addition, it might be beneficial to include downstream readouts such as cyclin levels or CDK activation that are more directly involved in cell cycle progression and can be deregulated in cancer [43, 44]. Our model attributes signalling differences between cell lines to an apparent feedback from MAPK signalling to IGFR and/or RAF. However, our model is too coarse-grained to distinguish feedback regulation from other, potentially non-linear mechanisms of cross-talk. Ultimately, only mechanistic studies that e.g. include the use of cell lines that have mutant feedback will unveil if the feedbacks are responsible for the observed signalling phenotypes and inhibitor synergies. It should be also pointed out that our measurements only encompass one time point and that later dynamics of the MAPK pathway, such as transcriptional feedbacks, could also explain IMR32 resistance to vertical inhibition.

In summary, our results show that a quantitative understanding of differences in signalling networks can be very helpful to rationalize resistance, and to derive effective treatments. Future work should investigate if those feedback mechanisms exist in tumours *in vivo* and whether they could explain relapses. Our description of the wiring of the RAS/MAPK pathway in neuroblastoma will support the design of clinical trials using combinatorial treatments to prevent or overcome therapy resistance. In addition, the framework described here could be used to analyse signalling in tumours of individual patients While it will be technically challenging to assess signalling network responses in tumour patients, *ex vivo* cultures—so-called avatars—could be an option [45, 46]. We envision that learning features of robustness and vulnerability of tumours from signalling models on cell line panels might greatly reduce the required set of perturbations in those avatars that are sufficient to inform a model, and allow reliable stratification and prediction of treatment options.

## Materials and methods

### Cell lines

The neuroblastoma cell lines were obtained by courtesy of the Deubzer lab (Charité, Berlin) as part of the Terminate-NB consortium. The identity of the cell lines was confirmed with STR

profiling (see S2 Table), which were generated by Eurofins Cell Line Authentification Test and matched with the Cellosaurus STR similarity research tool [47]. All cell lines were grown in DMEM (Gibco, Life Technologies) with 3.5 g/L glucose (Sigma), 5 mM glutamine (Gibco, Life Technologies) and 10% FCS (Pan Biotech).

## Whole exome sequencing

DNA was extracted from the human neuroblastoma cell lines (see above), using the NucleoSpin Tissue kit (Macherey-Nagel) according to the manufacturer's protocol. From the DNA, libraries for whole-exome sequencing were prepared using the SureSelect Human All Exon V7 kit (Agilent) and the Illumina TruSeq Exome kit. The libraries were sequenced on Illumina HiSeq 4000 and Illumina NovaSeq 6000 sequencers. The read sequences and base quality scores were demultiplexed and stored in Fastq format using the Illumina bcl2fastq software v2.20. Adapter remnants and low-quality read ends were trimmed off using custom scripts. The quality of the sequence reads was assessed using the FastQC software. Reads were aligned to the human genome, assembly GRCh38, using the bwa mem software version 0.7.10 [48], and duplicate read alignments were removed using samblaster version 0.1.24 [49]. Copy-number alterations were determined using cnvkit version 0.1.24 [50]. Single-nucleotide variants (SNVs) were identified using strelka version 2.9.10 [51]. Afterwards, potential germline variants were filtered out by excluding all SNVs that had also been observed in at least 1% of samples in cohorts of healthy individuals, namely the 1000 Genomes Project [52] and the NHLBI GO Exome Sequencing Project [53] cohorts. The raw data are available on ENA under the accession number PRJEB40670.

## RNA sequencing

The cell lines were sequenced in 3 separate batches. The IMR32, KELLY, SKNAS, LAN6, NBEBC1 cell lines were prepared in triplicate, using a paired-end stranded protocol with 2x75 cycles per fragment and 2 more cell lines (NGP, SKNSH) were prepared in duplicate, using a paired-end stranded protocol with 2x150 cycles. Two more libraries (CHP212 and N206) were prepared using a paired-end stranded protocol with 2x75 cycles per fragment.

Raw sequencing data were rigorously checked for quality using FastQC. The reads were aligned to the human genome GRCh38 (without patches or haplotypes) and the GENCODE transcript annotation set using the STAR aligner software [54]. The read counts per gene were obtained using the featurecounts [55] method from the subread software package. The raw data are available on ENA under the accession number PRJEB40670.

## Drug sensitivity assay

Cells grown for 1 day in full medium were treated with the indicated drugs in 4 different concentrations (0.1, 1, 10 and 100 $\mu$M Fig 1B) along with the corresponding DMSO controls on the same plate. The growth of the cells was tracked by phase contrast imaging for 72h with 4 images per well taken every 2h using the Incucyte Zoom instrument (Essen BioScience) and the confluency estimated using the Incucyte Zoom Analysis software (Essen BioScience). The growth rate was estimated with a linear fit on the log-transformed confluency, and the IC50 was determined by fitting a sigmoid of the form:

$$V = \frac{1}{1 + \exp(-\log(C) + IC50) \times S}$$

to normalised growth rates (implemented in https://github.com/MathurinD/drugResistance). $V$ is the growth rate relative to DMSO control, $C$ is the concentration and the parameters $IC50$

and slope $S$ are fitted. See S1 Table for the fitted parameters and S2 File for the raw data and analysis scripts, as well as S18 and S19 Figs for example images.

## Synergy estimation

For the synergy assay, cells seeded the day before were treated with different concentrations of AZD6244 (0.1, 1, 10, 30 and 50 $\mu$M, Selleck Chemicals) in combination with NVP-AEW541 (0.1, 0.3, 1, 3 and 10 $\mu$M, Cayman Chemical) or LY3009120 (0.1, 0.3, 1, 3 and 15 $\mu$M, Selleck Chemicals). The synergy scores were determined using the R package synergyfinder [56] with the relative growth rates thresholded between 0 and 1 as input (0 meaning no growth or cell death and 1 meaning growth as fast as the DMSO control).

## Perturbation assay

Cells were seeded in 24 well plates and grown for 2 days in full medium followed by 24h in FCS-free medium before treatment with the same concentrations of ligands and inhibitors.

All inhibitors were dissolved in DMSO and cells were treated for 90 minutes at the following concentrations: GDC0941 (1 $\mu$M, Selleck Chemicals), AZD6244/Selumetinib (10 $\mu$M, Selleck Chemicals), MK2206 2HCl (10 $\mu$M, Selleck Chemicals), Rapamycin (10 $\mu$M, Selleck Chemicals), Sorafenib (10 $\mu$M, Selleck Chemicals), GS-4997 (10 $\mu$M, Selleck Chemicals) and TAE684 (10 $\mu$M, Selleck Chemicals).

The cells were treated for 30 minutes (60 minutes after inhibitor treatment) with ligands in a 0,1% PBS/BSA carrier solution at the following concentrations: EGF (25 ng/mL, Peprotech), PDGF (10 ng/mL, Peprotech), NGF (50 ng/mL, Peprotech) and IGF1 (100 ng/mL, Peprotech).

The cells were then lysed using BioRad Bio-Plex Cell Lysis Kit and measured using the Bio-Plex MAGPIX Multiplex Reader with a custom kit from ProtAtOnce with analytes p-cJUN (S63), p-p38 (T180/Y182), p-AKT (S473), p-ERK1/2 (T202/Y204,T185/Y187), p-MEK1 (S217/S221), p-S6K (T389) and p-RSK1 (S380). The p-RSK1 (S380) readout was discarded because of a low dynamic range.

The same procedure and analytes were used for the other perturbation assays in this paper. Refer to the main text for the exact inhibitors and concentrations used for each experiment.

## Signalling models

The model for each cell line was fitted separately from the corresponding perturbation data with the *createModel* function from the R package STASNet [21]. STASNet implements the variation of Modular Response Analysis (MRA) described in [13] and [21] that implements a dual effect of inhibitors as both a negative stimulus and a disruption of signal propagation. Under the hypothesis of pseudo-steady-state and locally linear dependencies between nodes, MRA models the response to a perturbation as

$$R = -\tilde{r}^k * S \tag{1}$$

where $R_{ij}$ is the global response of node $j$ after perturbation of node $i$, $\tilde{r}_{ij}^k$ is the local response of node $j$ after perturbation of node $i$ taking into account the effect of inhibition of node $k$, and $S_{ik}$ is the sensitivity of node $i$ to perturbation $k$. The pAKT readout was systematically removed if AKT inhibition was present because the AKT inhibitor MK2206 blocks AKT autophosphorylation [57], i.e acts upstream of the AKT node, while STASNet expects inhibitors to act downstream of their annotated target.

We designed a literature network consisting of the MAPK and PI3K/AKT signalling pathway as annotated in KEGG (https://www.genome.jp/kegg/pathway/hsa/hsa04010.html and

https://www.genome.jp/kegg-bin/show_pathway?hsa04151) with intermediate nodes suppressed, the addition of the well documented ERK->RAF feedback and all receptors corresponding to RTK. Each cell line was fitted first on the literature network, then we extended the networks independently using a greedy hill climbing approach, until no significant link could be added. We then performed successive rounds of reduction to identify the non-significant links. Most removed links relate to receptor connections. Only three connections not related to receptors were removed during this procedure, each for one cell line only. To facilitate model comparison, these links were ultimately retained in the model, as otherwise the model parameters would not be comparable. Those models with final topology yielded similar values for the inhibition parameters so we generated new models with those parameters fixed to the mean value across all 6 models and re-fitted each cell line with inhibitor values fixed. With this fitting strategy the links between models became directly comparable as the non identifiability induced by the inhibitor parameters was removed (Fig 3A). The high pMEK responder cell line models were fitted using the same procedure.

## Western blot

Cells were grown to confluency for 3 days in full medium and treated with AEW541 $10\mu M$ and/or AZD6244 $10\mu M$ or control DMSO for 4h then lysed using BioRad Bio-Plex Cell Lysis Kit. The lysates were run for 3h at a constant 45 mA in 10% acrylamid gels and blotted for 45 minutes at 400 mA on nitrocellulose. The membranes were stained for total protein using Pierce Reversible Protein Stain (Thermofischer 24580) and blocked for 30 minutes in 1:1 PBS: Odyssey blocking buffer. The primary antibodies were incubated overnight at 4C one at a time and the corresponding secondary during the following day for 2h at room temperature in 1:1 PBST/Odyssey. We used the following primary antibodies: pIGF1R beta$^{Y1135/Y1136}$ 1:1000 (CST 3024), pAKT $^{S473}$ 1:2000 (CST 4060), total MYCN 1:200 (Santa Cruz sc-53993) and pMEK$^{S217/S221}$ 1:1000 (CST 9154).

## TMT (phospho-)proteomics

For the proteomics and phosphoproteomics cells were grown to confluency for 3 days in full medium and treated with AEW541 $10\mu M$ and/or AZD6244 $10\mu M$ or control DMSO for 4h.

We used an adapted version of the TMT workflow [28]: samples were reduced, alkylated and digested with a combination of LysC (Wako) and Trypsin (Promega) using the the single-pot, solid-phase-enhanced sample preparation [58]. For each sample, an equal amount of peptide was then chemically labelled with TMTpro reagents [29]. Samples were randomly assigned to one of the first 15 TMT channels, while the 16th channel was composed of a superset of all the samples to allow multi-plex normalisation. Equal amounts of the labelling reactions were combined in two TMT16 plexes, desalted via SepPak columns (Waters) and fractionated via high-pH fractionation [59] on a 96 minutes gradient from 3 to 55% acetonitrile in 5 mM ammonium formate, each fraction collected for 1 minute then combined into 24 fractions. From each fraction, an aliquot was used to measure the total proteome while the remaining peptides were combined into 12 fractions and used as input for an immobilised metal affinity chromatography using an Agilent Bravo system. For the total proteome analysis, peptides were on-line fractionated on a multi-step gradient from 0 to 55% acetonitrile in 0.1% formic acid prior injection in a QExactive HF-x mass spectrometer. Samples were acquired using a data dependent acquisition strategy with MS1 scans from 350 to 1500 m/z at a resolution of 60 000 (measured at 200 m/z), maximum injection time (IT) of 10 ms and an automatic gain control (AGC) target value of $3 \times 10^6$. The top 20 most intense precursor ions with charges from +2 to +6 were selected for fragmentation with an isolation window of 0.7 m/z. Fragmentation was

done in an HCD cell with a normalised collision energy of 30% and analysed in the detector with a resolution of 45 000 (200 m/z), AGC target value of $10^5$, maximum IT of 86 ms. We used the same parameters for phosphoproteome analysis with the exception of MS2 maximum IT that was set to 240 ms.

The acquired raw files were analysed using MaxQuant v1.6.10.43 [60], with TMTpro tags manually added as fixed modifications and used for quantitation. The correction factors for purity of isotopic labels was set according to vendor specification and minimum reporter precursor intensity fraction was set to 0.5. The resulting protein groups were filtered for potential protein contaminants, protein groups only identified via peptides decorated with modification or hits in the pseudo-reverse database used for FDR control. The resulting intensities of each sample channel were normalised to the intensity of the 16th reference channel, then median-centered and normalised according to the median-absolute deviation. Identified phosphopeptides were similarly filtered, with the exception of filtering based on modified sites, and normalised using the same strategy.

Differentially expressed phosphopeptides were called using the *limma* package [61] with a false discovery rate of 0.05 on treatment minus control contrasts. Synergies were computed using a contrast fit of the combination minus the sum of single treatments. Kinase substrate activity was implemented in R using the ratio of the mean z-score as described in [31] and computed for kinase-substrate sets from PhosphoSitePlus [62]. The normalised intensities and scripts used for the analysis can be found at https://itbgit.biologie.hu-berlin.de/dorel/phosphoproteomics_tnb_perturbations.

## Supporting information

**S1 Fig. Annotated IC50 of all measured drugs and cell lines with MYCN and TERT expression information.**
(PDF)

**S2 Fig. IC50 and mutations.** t-test comparison of the IC50 in mutant (Mut) versus wild type (WT) for RAS/P53 and associated genes with mutation frequency between 30% and 70% in our panel.
(PDF)

**S3 Fig. IC50 and gene expression.** Top correlation between IC50 and mRNA transcript per million for the 1000 most variable genes (top, adjusted p>0.93) and GO signal transduction genes (bottom, adjusted p>0.94).
(PDF)

**S4 Fig. PCA on the 1000 most variable genes.** Principal component analysis of the 1000 most variable genes. All components up to the first one explaining less than 10% of the variance are shown.
(PDF)

**S5 Fig. PCA on the signal transduction genes.** Principal component analysis of the 5262 signal transduction genes. All components up to the first one explaining less than 10% of the variance are shown.
(PDF)

**S6 Fig. Selected gene-drug correlations.** Correlation of NF1 expression with AZD6244 IC50 and ALK expression with TAE684 IC50.
(PDF)

**S7 Fig. Receptors RNA expression (TPM).**
(PDF)

**S8 Fig. Adaptors and ERBB receptor family RNA expression (TPM).**
(PDF)

**S9 Fig. Perturbation data PCA.** Pair-plot of the principal components from the perturbation in 2. All components up to the first one explaining less than 10% of the variance are shown.
(PDF)

**S10 Fig. Perturbation PCA loadings.** Main loadings in the first 3 principal components of the perturbation data PCA. Colors correspond to the component for which the condition has the highest absolute weight. Table indicates the weight for the top 10 conditions of the first 3 principal components.
(PDF)

**S11 Fig. Models qq-plots.** Quantile-quantile plots of the initial models using the (A) literature topology and (B) the final model after extension.
(PDF)

**S12 Fig. Correlation of signaling and receptor expression.** Correlation between the fitted path value from ligands to readouts and the expression of the matching receptor or receptor family. IGFRsum and PDGFRsum are the sum of the isoforms expression for IGFR and PDGFR respectively.
(PDF)

**S13 Fig. AZD6244 perturbation versus IC50.** Linear model fit of AZD6244 IC50 response to perturbations including AZD6244. Points are independent replicates, n = 2.
(PDF)

**S14 Fig. Differential phosphopeptides.** Differentially measured phosphopeptides in IMR32 and N206 after 4h inhibition (FDR < 0.05, n = 3) classified by treatment(s) where the phosphosite is differentially expressed.
(PDF)

**S15 Fig. KEGG enrichment of phosphoproteomics.** KEGG enrichment of unique genes corresponding to phosphopeptides differentially expressed after MEKi, IGFRi or MEKi+IGFRi treatment in (A) IMR32, (B) N206 or (C) both strictly. Enrichment was computed using the R package enrichKEGG.
(PDF)

**S16 Fig. Loewe synergy.** Loewe synergy for the combinations of AZD6244 with (A) AEW541 or (B) LY3009120 shown in 6B. Synergy scores were computed with the R package synergyfinder. Positive scores indicate synergy, negative scores indicate antagonism.
(PDF)

**S17 Fig. Viability to combination treatments.** Relative viability of IMR32, KELLY and N206 after treatment with AZD6244, AEW541 and RO5126766 alone or in combination. Confluency was tracked for 72h using the Incucyte Zoom. Growth rate was fitted to the confluency curve and normalised to the average growth rate of the corresponding DMSO controls. black crosses indicate the mean value for each cell line for the corresponding treatment.
(PDF)

**S18 Fig. Incucyte CHP212.** Incucyte image of the AZD6244-sensitive cell line CHP212 immediately after and 72h after DMSO or AZD6244 treatment.
(PDF)

**S19 Fig. Incucyte IMR32.** Incucyte image of the AZD6244-resistant cell line IMR32 immediately after and 72h after DMSO or AZD6244 treatment.
(PDF)

**S1 Table. IC50 data.**
(CSV)

**S2 Table. STR profiling results.**
(XLSX)

**S1 File. Sequencing data and related analysis scripts.**
(ZIP)

**S2 File. Dose response data and related analysis scripts.**
(ZIP)

**S3 File. Model data and fitting summary.**
(ZIP)

## Acknowledgments

We thank Aleixandria McGearey for technical assistance with preparing the whole-exome sequencing libraries, Martha Hergesekke for help in cell culture, as well as Jasmin Wünschel for providing the cell lines.

## Author Contributions

**Conceptualization:** Bertram Klinger, Nils Blüthgen.

**Data curation:** Mathurin Dorel, Tommaso Mari, Joern Toedling, Eric Blanc, Clemens Messerschmidt, Michal Nadler-Holly, Matthias Ziehm, Falk Hertwig.

**Funding acquisition:** Angelika Eggert, Matthias Selbach, Nils Blüthgen.

**Investigation:** Mathurin Dorel, Tommaso Mari.

**Methodology:** Bertram Klinger, Anja Sieber.

**Project administration:** Nils Blüthgen.

**Resources:** Matthias Ziehm, Falk Hertwig, Johannes H. Schulte.

**Software:** Mathurin Dorel, Bertram Klinger, Nils Blüthgen.

**Supervision:** Bertram Klinger, Dieter Beule, Angelika Eggert, Johannes H. Schulte, Matthias Selbach, Nils Blüthgen.

**Visualization:** Mathurin Dorel.

**Writing – original draft:** Mathurin Dorel.

**Writing – review & editing:** Bertram Klinger, Nils Blüthgen.

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
