## [Decision Letter · Decision Letter 0]

30 Jul 2021

Dear Dr. Blüthgen,

Thank you very much for submitting your manuscript "Neuroblastoma signalling models unveil combination therapies targeting feedback-mediated resistance" for consideration at PLOS Computational Biology. As with all papers reviewed by the journal, your manuscript was reviewed by members of the editorial board and by several independent reviewers. The reviewers appreciated the attention to an important topic. Based on the reviews, we are likely to accept this manuscript for publication, providing that you modify the manuscript according to the review recommendations.

Sincerely,

Inna Lavrik

Associate Editor

PLOS Computational Biology

Mark Alber

Deputy Editor

PLOS Computational Biology

[LINK]

Reviewer's Responses to Questions

**Comments to the Authors:**

Reviewer #1: In this paper the authors investigate mechanisms of drug resistance in a panel of high risk neuroblastoma cells focusing on the ERK pathway that is frequently altered in high risk neuroblastoma. They use a combination of experimental perturbation data and an elegant mathematical method to reconstruct signaling networks in order to decipher molecular mechanisms of drug resistance. Surprisingly, rather that linked to genetic mutations drug resistance seems mediated by differential network topologies in particular different feedback loops. While not entirely new, this finding is an important contribution to the growing case that network adaptations can be responsible for drug resistance in cancer and that we need to look beyond genetic mutations. Overall, the paper is well conceived, and the combination of experimental work and mathematical modelling is convincing. I am happy to recommend publication, but have a few comments that should be addressed.

Major comments

As negative feedback loops are ultimately made responsible for the differences in drug resistance, it would strengthen these conclusions by showing more evidence of their direct involvement. For instance, one would expect phosphorylation sites to change that mediate these feedback in a manner that correlates with the feedback strengths. These data seem to be there, but summarizing them in a figure would make the message clearer. Also, mutants that break the feedback could be used to show that is really the feedback that mediates drug resistance.

Fig. 6B. The authors claim a “strong synergistic effect of the combination of MEK and IGFR inhibitions on growth 291 in N206 but little in KELLY or IMR32.” This is not apparent from the figure. A formal measure of synergism should be used, such as the Chou-Talalay method or Loewe isoboles.

Fig. 6D. These data are not very convincing. The triple drug combination does not seem to be more potent than the dual combinations. This figure need error bars and a proper statistical comparison.

Minor comments

The MYN amplification status of the cell lines should be reported.

“A strong RAF-mediated feedback is a known resistance mechanism against MEK inhibitors (Friday et al, 2008; Fritsche-Guenther et al, 2011), …” should also cite Sturm et al. 2010.

“… a known nexus between AKT and MAPK signalling (?).” Please supply the citation.

MYCN phosphorylation on S62 has been linked to enhanced degradation using phosphosite mutants (https://doi.org/10.1016/j.ccr.2014.07.015). However, here the authors associate the de-phosphorylation of this residue with lower MYCN protein levels. Would be worth discussing or mitigating this claim as it is only based on a correlation.

Reviewer #2: The authors have used quantitative perturbation to study the resistance mechanisms in well characterized Neuroblastoma cell lines. They focused on a handful of ligands and signaling genes to build a prior knowledge network. They tested different drugs and combination to fit the network and identify cell-line specific mechanisms like negative feedbacks. They could even pinpoint a vertical inhibition to counter the feedback-mediated resistance in studied cell lines. All together this manuscript represents an important step in the evaluation of resistance mechanisms in cancer cell lines and might pave the way for sensitizing resistant tumors and improve patient outcome.

Nevertheless, the author should address the following comments before publication:

- To my understanding the authors have only try to extend the model, by adding edges between genes. Did they also try to remove edges? Although they are well characterized in the literature, I was wondering if they were all essential on if some could be discarded in one or many cell lines. Perhaps it would improve the fit.

- IC50 estimation in this study was done by monitoring cell growth with the Incucyte platform. As this instrument generates masks to calculate the confluency of cells, this can be biased by differences in cell morphology or patterns of cell growth. Did the authors consider whether the morphology of the cells was similar before and after drug treatment? It could be of profit to include, as supplementary data, some raw images of the cells at the beginning and end of the recording (72 h), at least for MEKi.

- Supplementary Figure2: Which method was used to calculate the correlation? As mutation data are qualitative (or at least Boolean), Pearson’s correlation is maybe not the best method to apply here. The authors should add more details about this and perhaps try other methods, like non parametric tests, to infer association between IC50 and mutations.

- Line 94-96: “To get insights into the underlying mechanisms of resistance to the MEK inhibitor AZD6244, we selected 6 neuroblastoma cells lines that represented the spectrum of sensitivity to MEK inhibition (sensitive: CHP212, LAN6; resistant: SKNAS, SKNSH, KELLY and IMR32)”, the period at the end of the sentence is missing.

- Page 9, line249: What does the (?) stand for?

- In figure 5 part C legend: there is a typo on phosphopeptides

- Supplementary Figure 3: Is it really adjusted p>0.95? In the text it is written 0.93 and 0.94 for 1000 most variable genes and GO signal transduction genes respectively. Please double check.

- Supplementary Figure 4,5,9: color code is missing

-Supplementary Figure 17: Gene-sets should be re-ordered according to the GeneRatio to better see the ranking.

**Have the authors made all data and (if applicable) computational code underlying the findings in their manuscript fully available?**

Reviewer #1: Yes

Reviewer #2: Yes

PLOS authors have the option to publish the peer review history of their article (what does this mean?). If published, this will include your full peer review and any attached files.

Reviewer #1: No

Reviewer #2: No

Figure Files:

Data Requirements:

Reproducibility:

References:

---

## [Editor Report · Decision Letter 1]

1 Oct 2021

Dear Dr. Blüthgen,

We are pleased to inform you that your manuscript 'Neuroblastoma signalling models unveil combination therapies targeting feedback-mediated resistance' has been provisionally accepted for publication in PLOS Computational Biology.

Best regards,

Inna Lavrik

Associate Editor

PLOS Computational Biology

Mark Alber

Deputy Editor

PLOS Computational Biology

---

## [Editor Report · Acceptance letter]

28 Oct 2021

PCOMPBIOL-D-21-01059R1 

Neuroblastoma signalling models unveil combination therapies targeting feedback-mediated resistance

Dear Dr Blüthgen,

I am pleased to inform you that your manuscript has been formally accepted for publication in PLOS Computational Biology. Your manuscript is now with our production department and you will be notified of the publication date in due course.

With kind regards,

Livia Horvath
